# Mobile game addiction and its association with musculoskeletal pain among students: A cross-sectional study

Sohel Ahmed[1,2]*, Asir John Samuel[3], Arushi Mishra[4], Md Saifur Rahman[5], Md. Ariful Islam[6], Md. Rashaduzzaman[7], Shankar Kumar Roy[8], Rahemun Akter[1], Mohammad Jahirul Islam[9]

1 Ahmed Physiotherapy & Research Center, Dhaka, Bangladesh, 2 Directorate of Students' Welfare, Bangladesh University of Engineering and Technology, Dhaka, Bangladesh, 3 Yenepoya Physiotherapy College, Yenepoya (Deemed-to-be University), Naringana, Karnataka, India, 4 Department of Physiotherapy, Maharishi Markandeshwar Medical College and Hospital, Solan, Himachal Pradesh, India, 5 Department of Physiotherapy and Rehabilitation, Jashore University of Science and Technology, Jashore, Bangladesh, 6 Department of Physiotherapy, Zaman Mordan Hospital, Sherpur, Bangladesh, 7 Bangabandhu Sheikh Mujib Medical University, Dhaka, Bangladesh, 8 Department of Orthopedics, MAG Osmani Medical College Hospital, Sylhet, Bangladesh, 9 Department of Physical Medicine and Rehabilitation, MAG Osmani Medical College Hospital, Sylhet, Bangladesh

☯ These authors contributed equally to this work.
* ptsohel@gmail.com

**Data Availability Statement:** All relevant data are within the paper and its Supporting Information file.

## Abstract

### Background

The purpose of this study was to ascertain whether there is a difference in musculoskeletal pain between those who are addicted to mobile games and those who are not, to ascertain the association between mobile game addiction and socio-demographic variables, and to ascertain the pain predictor for mobile game addiction on different musculoskeletal regions.

### Methods

There were 840 students in all, both males and females, in this cross-sectional survey from three distinct Bangladeshi institutions. The Nordic Musculoskeletal Discomfort Questionnaire, the Gaming Addiction Scale, and the demographic data form were distributed to the participants. The data were analyzed using the Chi-square test and descriptive statistics. Binary logistic regression was used to find the predicted risk factor for mobile gaming addiction.

### Results

Musculoskeletal pain affects 52.1% of participants in some part of the body. Males have a 2.01-fold higher likelihood of developing gaming addiction compared to females. Those who are addicted to mobile games experience a higher occurrence of pain in the neck, upper back, elbows, and wrist and hands with a odds ratio of (OR 2.84, 95% CI: 1.49–5.36; p = 0.016), (OR 3.75, 95% CI 1.97–7.12; p = <0.001), (OR 3.38, 95% CI 1.34–8.50; p = 0.010), and (OR 2.14, 95% CI 1.00–4.57; p = 0.049) respectively.

**Funding:** The author(s) received no specific funding for this work.

**Competing interests:** The authors have declared that no competing interests exist.

## Conclusion

These results demonstrate that mobile gaming addiction raises students' risk of musculo-skeletal discomfort. Two-three times higher risk of developing pain in the neck, upper back, elbows, and wrist and hands among mobile game addicts.

## Introduction

The smartphone is one of the most innovative and modern technologies of the 21st century because it extends its amazing range of digitally mediated channels and acts as a polymedia [1]. Communication, entertainment, sports, and education are all possible on a mobile phone [2]. In recent years, there has been a drastic increase in the trend of playing mobile games among the young generation, which can become a serious public health issue [3]. Though playing mobile games for a short time can positively affect mental and physical health [4], playing electronic games for a prolonged period may adversely impact health, social relationships, and physical activity [5].

The World Health Organization classified mobile game addiction as a mental disorder in the 11th revision of the International Classification of Diseases (ICD) in 2018 [6]. Excessive gaming addiction causes lower life satisfaction and a higher level of anxiety and depression [7]. Daily computer use of 2 hours or more increases the risk of musculoskeletal symptoms among adolescent computer users, as reported by Hakala, P.T et al [8]. Continuously playing a smart-phone game for 30 minutes may result in chronic muscle injury on the thumbs [9]. Everyday online gaming for more than five hours a day was associated with depressive, psychometric, and musculoskeletal symptoms among adolescents who engage in online gaming [10]. More than 30% of elite e-sports players reported headaches and rhinitis [11].

Smartphone use in poor and awkward posture causes additional stress on the muscles, which has a number of negative effects on the body [12, 13]. Awkward posture impairs physio-logical function, disturbs the autonomic nervous system, results in various problems in daily life, and has an impact on the musculoskeletal and visual systems. As a result, the likelihood of musculoskeletal pain and discomfort will increase [14]. Sitting in front of digital devices with bad posture for long periods of time is responsible for muscle pain [15]. A Turkish research study [16] discussed the physical pain and mental health issues experienced by e-sports com-petitors. The literature on mobile game addiction's impact on students' musculoskeletal pain is scarce. So, the purpose of this study was to find out the association between mobile game addiction and the socio-demographic variables of the participants, to determine if there is a difference in musculoskeletal pain between those who are addicted to mobile games and those who are not, as well as the pain predictor for mobile game addiction in different musculoskele-tal regions among students.

## Methodology

From March to June 2022, students from various parts of Bangladesh participated in this cross-sectional research. The ethical review board of Institute of Physiotherapy, Rehabilitation and Research affiliated with Bangladesh Physiotherapy Association granted permission with the reference number BPA-IPRR/IRB/11/02/2022/71. The Bangladesh Medical Research Council's ethical guidelines and the Declaration of Helsinki (2013 revision) were rigorously observed in this research, which included human participants. A digital informed consent was

obtained from the participants and ascent was obtained from the legal guardian in case of minority prior to participate in this study. The S1 Checklist includes the Strengthening the Reporting of Observational Studies in Epidemiology (STROBE) guideline, which served as the basis for reporting the study.

## Study population

Students from three different institutions served as the population source. Regular high school (9th to 12th grade), and college students (diploma in medical technology, undergraduate, and postgraduate), and those who met the inclusion and exclusion criteria were included in this research.

## Subject selection criteria

a. **Inclusion criteria:** Regular students who own a smartphone themselves or through their parents were included in this study.

b. **Exclusion criteria:** This study excluded students who had any documented musculoskeletal diseases, a history of musculoskeletal surgeries, a disc condition, or a history of trauma. Prior to data collection, we created a checklist with yes-or-no questions to make sure those requirements were met.

## Sample size estimation and sampling methods

Using the method for single population proportions, the sample size was estimated under the assumptions of a 95% level of significance, a 3.5% margin of error, and a response distribution of 50% were used to calculate the sample size [17]. The following calculation, where n = sample size, p = prevalence, and d = margin of error, was used to determine the necessary sample size:

$$N = z^2 p(1-p)/d^2, n = (1.96)^2 x(0.5)(0.5)/(0.035)^2, = 784$$

By including 5% incomplete forms it required 823 samples, hence, 840 people made up the final sample size.

The research study used a stratified random sample approach to gather data. The process of stratification was predicated upon the categorization of individuals according to their academic attainment, including secondary education, upper secondary education, diploma programs, undergraduate studies, and postgraduate studies. The sample used in this study was obtained from three distinct educational institutions located in Bangladesh. The overall sample size was partitioned into five strata, and the allocation of 168 samples per stratum was established by the use of random sampling techniques.

## Survey development

### a. Socio-demographic variables

Demographic characteristics such as gender, age, weight, height, level of education, and use of personal devices were included. This study also includes mobile gaming details, including the following questions: How many hours per day (approximately) do you play games? How long have you been playing mobile games? How frequently do you buy mobile data to play games? How frequently do you play commercial games? How frequently do you spend money on online games? were included.

*b. Game addiction scale (GAS)*

At present, majority of the mobile phone users (school, college or university) have smartphone with the access of internet and spent time with internet-based mobile games [18]. Hence, it is more appropriate to evaluate the internet gaming addiction as mobile game addiction. The GAS consists of seven questions on a five-point Likert scale. Using a scale of 1 (never) to 5 (very often), the total score is between 7 and 25. A higher score indicates the high risk of gaming addiction. The scale is valid and reliable to measure game addiction among the adolescent population [19].

*c. Pain history*

Do you experience any pain in your body in the past 7 days? Was used to evaluate the recent history of musculoskeletal pain. If the subject replied "yes," the Nordic musculoskeletal disorder questionnaire was used to determine the location of the pain. Nine different locations on the body, including the neck, upper back, shoulder, elbow, wrist or hand, lower back, hip or thigh, knee, and ankle or foot, were listed on the questionnaire. Previous work has already confirmed the reliability and validity of the Nordic musculoskeletal questionnaire [20].

## Survey validation and administration

Since the tools we utilized had already been established, the survey questionnaire's draft was not subjected to content validation. The questionnaire's final draft was first tested on five representative populations to determine response times; the average time was less than 10 minutes. Responses to the survey were gathered using the Google Forms® platform. The information was gathered via a questionnaire that was given to the participants. The option necessary is included in every question on Google Forms to prevent incomplete submissions.

## Data analysis

The data were organized, cleaned up, and reviewed for internal consistency before being analyzed. Descriptive statistics were expressed as frequency, percentage, mean, and standard deviation. The Fisher's exact test or Pearson's Chi-square test were employed to determine whether there is a difference between addicted and non-addicted players of mobile games. Using odds ratios (OR) and a 95% confidence interval (CI), a binary logistic regression analysis was conducted to determine the predictor of pain in different body regions for mobile gaming addiction. The independent variables in this study were mobile game addiction and age, gender, weight, height, and the presence of pain in different body regions as predictor variables for mobile game addiction. The model's fitness was assessed using Hosmer-Lemshow's test and classification table. To see if there was multicollinearity between the independent variables, variance inflation factors (VIFs) with a threshold of VIF≤5.0 was used. In each test, the level of significance was fixed at <0.05. IBM SPSS Statistics for Windows 11, Version 27.0 (Armonk, NY: IBM Corp). was used for all data analysis.

## Results

A total of 840 participants—539 men, or 64.2%—and 301 women, or 35.8%—participated in the current study. The participants' mean ages, weights, and heights were correspondingly 20.95 ± 4.84 years, 56.73 ± 12.49 kg, and 150.10 ± 29.47 cm. 95.4 percent of the participants in this research are mobile phone owners. 19.9% of participants play offline games, 42.7% play online games, and 37.5% do not play mobile games. In this study, 4.3% of participants played mobile games for longer than four hours. 16.1% of participants report occasionally paying

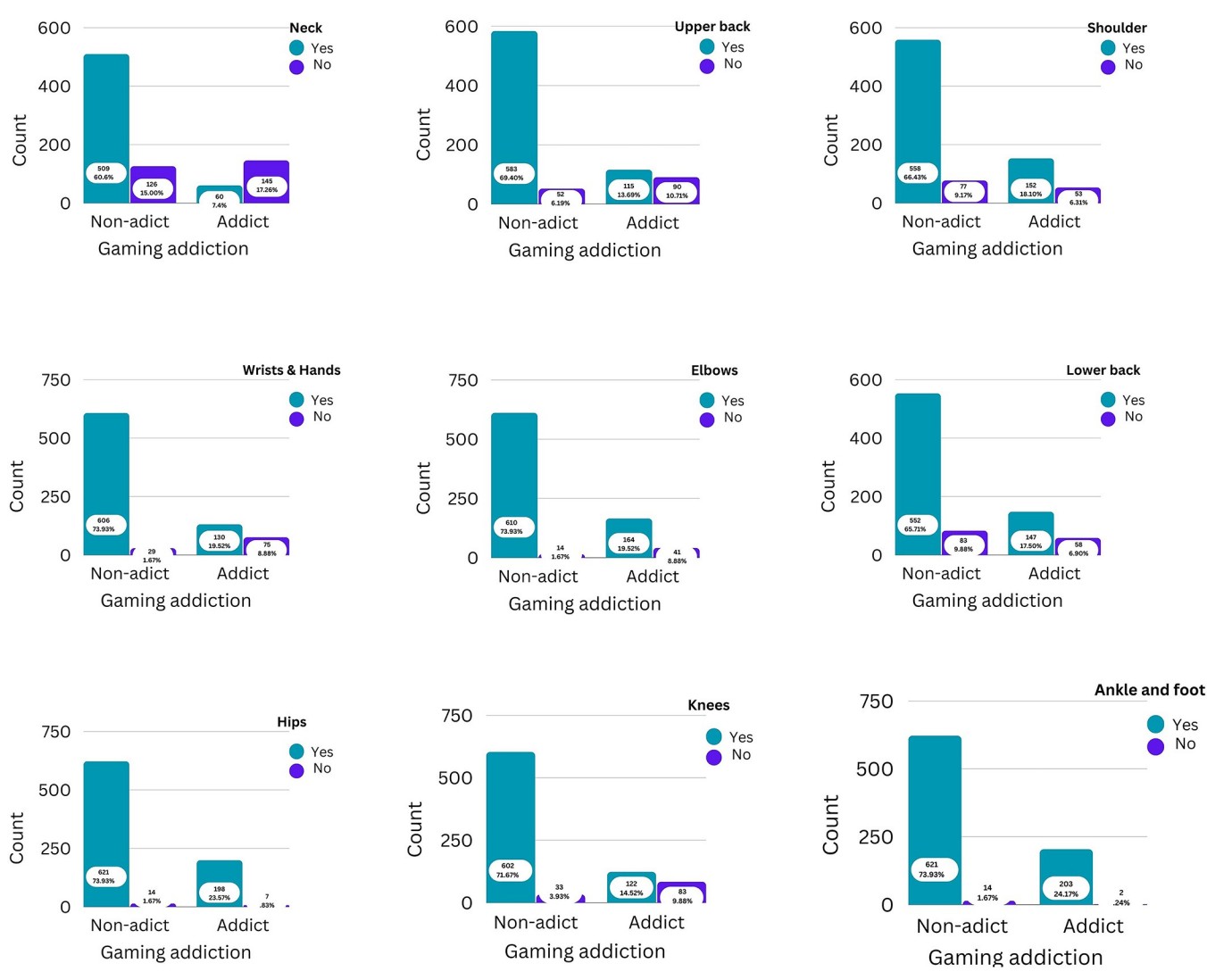

**Fig 1. Prevalence of musculoskeletal pain in various body regions.**

money to play online games. Among the participants, 42.5% do not have a mobile gaming addiction, 33.1% have a mild addiction, 23.6% have a moderate addiction, and 0.8% have a severe addiction. Musculoskeletal pain affects 52.1% of participants in some part of the body. Fig 1 demonstrates the prevalence of musculoskeletal pain in different body parts among mobile game addicts and non-addicts.

## Association between mobile game addiction and socio-demographic variables

Addiction to mobile games was significantly correlated with the socio-demographic characteristics of the participants, such as education level (p = 0.001) and device ownership (p = 0.001). Also, there is a significant relationship (p <0.001) between how long mobile games are played, how long they have been played (p <0.001), and what kinds of games are played (p <0.001). Details are presented in Table 1.

**Table 1. Socio-demographic variables of the study participants (n %).**

| Variables | Not addicted | Addicted | Total | P value |
|---|---|---|---|---|
| Gender | | | | |
| Male | 400 (63.0) | 139 (67.8) | 539 (64.2) | 0.212 |
| Female | 235 (37.0) | 66 (32.2) | 301 (35.8) | |
| Education level | | | | |
| Secondary | 128 (20.2) | 40 (19.5) | 168 (20.0) | <0.001 |
| Higher-secondary | 114 (18.0) | 54 (26.3) | 168 (20.0) | |
| Diploma | 146 (23.0) | 22 (10.7) | 168 (20.0) | |
| Undergraduate | 102 (16.1) | 67 (32.7) | 169 (20.1) | |
| Post-graduate | 145 (22.8) | 22 (10.7) | 167 (19.9) | |
| Ownership of devices | | | | |
| Yes | 598 (94.2) | 203 (99.0) | 801 (95.4) | 0.002** |
| No | 37 (5.8) | 2 (1.0) | 39 (4.6) | |
| How many hours each day do you play mobile games? | | | | |
| Not at all | 317 (49.9) | 0 (0) | 317 (37.7) | <0.001** |
| 0.1–1 hour | 129 (20.3) | 30 (14.6) | 159 (18.9) | |
| 1.1–4 hours | 176 (27.7) | 152 (74.1) | 328 (39.0) | |
| >4 hours | 13 (2.0) | 23 (11.2) | 36 (4.3) | |
| Source of internet | | | | |
| Wi-Fi | 163 (25.7) | 20 (9.8) | 183 (21.8) | <0.001 |
| Mobile data | 202 (31.8) | 115 (56.1) | 317 (37.7) | |
| Both | 270 (42.5) | 70 (34.1) | 340 (40.5) | |
| What type of mobile game do you usually play? | | | | |
| I don't play mobile games | 315 (49.6) | 0 (0) | 315 (37.5) | <0.001** |
| Online single player | 55 (8.7) | 48 (23.4) | 103 (12.3) | |
| Online multiplayer | 156 (24.6) | 99 (48.3) | 255 (30.4) | |
| Offline game | 109 (17.2) | 58 (28.3) | 167 (19.9) | |
| How long have you been playing mobile games? | | | | |
| I don't play mobile games | 310 (48.8) | 0 (0) | 310 (36.9) | <0.001** |
| <1 year | 84 (13.2) | 21 (10.2) | 105 (12.5) | |
| 1–2 years | 116 (18.3) | 46 (22.4) | 162 (19.3) | |
| 3–5 years | 72 (11.3) | 76 (37.1) | 148 (17.6) | |
| >5 years | 53 (8.3) | 62 (30.2) | 115 (13.7) | |
| How often do you buy mobile data to play games? | | | | |
| Never | 439 (69.1) | 10 (4.9) | 449 (53.5) | <0.001 |
| Rarely | 64 (10.1) | 26 (12.7) | 90 (10.7) | |
| Sometimes | 101 (15.9) | 68 (33.2) | 169 (20.1) | |
| Often | 17 (2.7) | 71 (34.6) | 88 (10.5) | |
| Very often | 14 (2.2) | 30 (14.6) | 44 (5.2) | |
| How often do you play commercial games? | | | | |
| Never | 527 (83.0) | 46 (22.4) | 573 (68.2) | <0.001** |
| Rarely | 51 (8.0) | 60 (29.3) | 111 (13.2) | |
| Sometimes | 49 (7.7) | 65 (31.7) | 114 (13.6) | |
| Often | 6 (0.9) | 15 (7.3) | 21 (2.5) | |
| Very often | 2 (0.3) | 19 (9.3) | 21 (2.5) | |
| How often do you spend money on online games? | | | | |

(*Continued*)

**Table 1.** (Continued)

| Variables | Not addicted | Addicted | Total | P value |
|---|---|---|---|---|
| Never | 534 (84.1) | 65 (31.7) | 599 (71.3) | <0.001** |
| Rarely | 46 (7.2) | 59 (28.8) | 105 (12.5) | |
| Sometimes | 51 (8.0) | 67 (32.7) | 118 (14.0) | |
| Often | 2 (0.3) | 10 (4.9) | 12 (1.4) | |
| Very often | 2 (.03) | 4 (2.0) | 6 (0.7) | |
| Do you have pain in your body? | | | | |
| Yes | 263 (41.4) | 175 (85.4) | 438 (52.1) | <0.001 |
| No | 372 (58.6) | 30 (14.6) | 402 (47.9) | |

## Association between different musculoskeletal pain regions among the mobile game addicts and non-addicts

Among addicts and non-addicts, mobile game addiction is significantly associated with different musculoskeletal pain regions: neck (p <0.001), shoulders (p <0.001), upper back (p <0.001), elbows (p <0.001), wrists and hands (p <0.001), lower back (p <0.001), and knees (p <0.001). No significant association was obtained for the areas of the hips, ankles, and feet. Phi and Crammer's V demonstrate moderate association in the neck, upper back, wrist, hands, and knees. Details are presented in Table 2.

## Mobile game addiction as a predictor of musculoskeletal pain

Males have a 2.01-fold higher likelihood of developing gaming addiction compared to females (OR p = 0.016, 95% CI: 1.139–3.532). Individuals who are addicted to mobile games are susceptible to the potential development of physical discomfort. Compared to individuals who are not addicted to mobile gaming, those who are addicted experience a higher occurrence of pain in the neck, upper back, lower back, elbows, and wrist and hands, with odds ratios of (OR 2.84, 95% CI 1.49–5.36; p = <0.001), (OR 3.75, 95% CI 1.97–7.12; p = <0.001), (OR 1.78, 95% CI 0.93–3.40; p = 0.083), (OR 3.38, 95% CI 1.34–8.50; p = 0.010), and (OR 2.14, 95% CI 1.00–4.57; p = 0.049) respectively. Table 3 displays the binary logistic regression model, which serves as a predictive tool for assessing the relationship between pain and mobile gaming addiction.

**Table 2.** Association between different musculoskeletal pain regions among the mobile game addicts and non-addicts.

| Association b/w status of gaming addiction with pain region | Chi-square | | Phi-Crammer's V | | |
|---|---|---|---|---|---|
| | Statistic | p-value | Phi | Crammer's V | p-value |
| Neck | 183.6 | <0.001 | 0.468 | 0.468 | <0.001 |
| Shoulders | 22.3 | <0.001 | 0.163 | 0.163 | <0.001 |
| Upper back | 140.7 | <0.001 | 0.409 | 0.409 | <0.001 |
| Elbows | 80.2 | <0.001 | 0.309 | 0.309 | <0.001 |
| Wrist and hand | 146.5 | <0.001 | 0.418 | 0.418 | <0.001 |
| Lower back | 25.7 | <0.001 | 0.175 | 0.175 | <0.001 |
| Hips | 0.931 | 0.335 | 0.033 | 0.033 | 0.335 |
| Knees | 162.2 | <0.001 | 0.439 | 0.439 | <0.001 |
| Ankles and foots | 1.253 | 0.263 | 0.039 | 0.039 | 0.263 |

**Note:** Phi Cramer's v interpretation: 0.0 to 0.1 –Negligible association; 0.1 to 0.2 –weak association; 0.2 to 0.4 –moderate association; 0.4 to 0.6 –relatively strong; 0.6 to 0.8 –strong association; 0.8 to 1.0 –very strong association

**Table 3. Binary logistic regression; a predictor of pain for mobile game addiction.**

| Variables | | Adjusted OR | SE | 95% CI | | P value |
|---|---|---|---|---|---|---|
| | | | | Lower | Upper | |
| Age | | 0.969 | .030 | 0.915 | 1.027 | 0.290 |
| Gender | | 2.005 | .289 | 1.139 | 3.532 | 0.016 |
| Weight | | 0.800 | .036 | 0.746 | 0.858 | <0.001 |
| Height | | 1.097 | .014 | 1.068 | 1.128 | <0.001 |
| Body pain | Neck | 2.835 | .325 | 1.499 | 5.362 | 0.001 |
| | Shoulders | 0.650 | .356 | 0.324 | 1.306 | 0.227 |
| | Upper back | 3.746 | .328 | 1.971 | 7.119 | <0.001 |
| | Elbows | 3.378 | .471 | 1.343 | 8.499 | 0.010 |
| | Wrists and hands | 2.141 | .388 | 1.001 | 4.579 | 0.049 |
| | Lower back | 1.777 | .331 | 0.928 | 3.402 | 0.083 |
| | Hips | 0.302 | .708 | 0.075 | 1.209 | 0.091 |
| | Knees | 1.488 | .404 | 0.675 | 3.282 | 0.325 |
| | Ankle and foots | 0.508 | .841 | 0.098 | 2.640 | 0.420 |

## Discussion

This study, the first of its kind to be carried out in Bangladesh, examines the association between mobile gaming addiction and musculoskeletal pain in several body regions. The results of this study reported that 24.4% of participants were moderately to severely addicted to mobile gaming; nevertheless, musculoskeletal pain is more common; 52.1% of individuals report having musculoskeletal pain in some area of the body, and mobile gaming addiction is significantly associated with different musculoskeletal pain sites such as the neck, shoulders, upper back, elbows, wrists, hands, lower back, and knees.

According to the participants in our study, there is a significant association between mobile gaming addiction and complaints of musculoskeletal discomfort. Earlier studies also came to similar conclusions, as Cankurtaran et al. investigated the effects of digital game addiction on the musculoskeletal system of secondary school students and discovered a significant correlation between game addiction and wrist, upper back, and low back pain [15]. Internet gaming doubles the likelihood that adolescents may experience musculoskeletal problems (OR 2.494, 95% CI 1.598–3.892) [10]. Online gaming addiction alters body posture [21] and raises the likelihood of musculoskeletal system pain [16] which is similar to the findings of our study.

Males were more addicted to mobile games compared to females in this present study. Males reported a double chance of developing musculoskeletal pain compared to females. Earlier studies also found similar results, as Cankurtaran et al. reported that males were more addicted to digital games [15]. A study of 1000 healthy children found that male students are more addicted to digital games than female students [15]. In another study, male participants in the research were more addicted to video games than female participants, according to a 2015 study that involved 678 students [22].

In our study, the prevalence of musculoskeletal discomfort was much greater among mobile gaming addicts. Video game play has been shown to have a deleterious influence on the musculoskeletal system, according to a systematic review of musculoskeletal diseases among gamers. According to the survey, the neck, shoulder, and back were the areas of the body that complained the most. Similar to our investigation, another study also found increased odds ratios for the relationship between playing video games and the development of musculoskeletal problems [23]. Another study by Ahmed et al. found that smartphone addiction was substantially linked to discomfort in the study participants' necks, shoulders, elbows, and wrists

[12]. 42.6% of e-sports athletes reported musculoskeletal pain within the past week, with the most prevalent pain areas among e-sports players being their backs, necks, and shoulders [24].

This study reported a significant difference among mobile game addicts and non-addicts in mobile gaming time, types of mobile games played, playing commercial games, spending money on mobile games, and the educational level of the participants. Playing video games for ≥3 hours in a day was associated with musculoskeletal pain and reported a 5.59 times higher odds ratio than those who played <1 hour in a day [25]. Males, students who did not do well in school, and those who enjoyed multiplayer online games all had a much higher risk of becoming addicted to gaming. Gaming addiction was significantly connected with the average amount of time spent each week playing games, the frequency with which people spend money playing games, and the length of time they play [26].

### Strength, limitations and future recommendations

Data were gathered using a probability sampling approach and reliable data collection tools, which helps reduce bias. Face-to-face interviews using a structured questionnaire serve to improve the accuracy and validity of the data by decreasing non-response and any misclassification bias. The generalizability of the results may be improved by including individuals from various locations. Yet, there are certain limitations to this study as well. A causal association between the related variables could not be confirmed due to the cross-sectional nature of the research methodology used in this study. Participants provided their own information, which carries the potential for recollection bias. As, the student population have sufficient proficiency in English to understand the meaning of each question, we do not find any practical difficulties to use it GAS in the study. To ascertain the real situation of mobile gaming addiction among students, a cross-sectional study should be conducted across the country. This study will be crucial in educating students about the negative impacts of mobile gaming addiction and its consequences. This study will also help students understand the harmful effects of mobile game addiction on the musculoskeletal system and reduce the risk of developing musculoskeletal pain.

### Conclusion

These results demonstrate that mobile gaming addiction raises students' risk of musculoskeletal discomfort. Mobile game addiction is significantly associated with different musculoskeletal pain regions, including the neck, shoulders, upper back, elbows, wrists, and hands, lower back, and knees. Mobile game players should consider these findings with an active lifestyle at the forefront.

### Supporting information

**S1 Checklist.**
(DOCX)

### Acknowledgments

The authors are thankful to the participants for providing the information used to conduct the study. In addition, we would like to express our sincere thanks to Yenepoya (Deemed to be University), Deralakatte, Mangalore, Karnataka, India for the statistical support provided by offering IBM SPSS Statistics for Windows 11, Version 27.0 (Armonk, NY: IBM Corp).

## Author Contributions

**Conceptualization:** Sohel Ahmed, Md Saifur Rahman, Md. Ariful Islam, Md. Rashaduzzaman, Shankar Kumar Roy, Rahemun Akter, Mohammad Jahirul Islam.

**Data curation:** Arushi Mishra, Md Saifur Rahman, Md. Ariful Islam, Md. Rashaduzzaman, Mohammad Jahirul Islam.

**Formal analysis:** Sohel Ahmed, Asir John Samuel, Arushi Mishra.

**Investigation:** Sohel Ahmed, Arushi Mishra, Md Saifur Rahman, Md. Ariful Islam, Md. Rashaduzzaman.

**Methodology:** Sohel Ahmed, Asir John Samuel, Rahemun Akter, Mohammad Jahirul Islam.

**Project administration:** Md Saifur Rahman, Md. Ariful Islam, Md. Rashaduzzaman, Shankar Kumar Roy, Mohammad Jahirul Islam.

**Software:** Sohel Ahmed, Asir John Samuel, Md Saifur Rahman, Mohammad Jahirul Islam.

**Supervision:** Asir John Samuel, Shankar Kumar Roy.

**Validation:** Sohel Ahmed, Asir John Samuel, Arushi Mishra, Md Saifur Rahman, Shankar Kumar Roy, Rahemun Akter, Mohammad Jahirul Islam.

**Visualization:** Sohel Ahmed, Asir John Samuel, Arushi Mishra, Md Saifur Rahman, Shankar Kumar Roy, Rahemun Akter, Mohammad Jahirul Islam.

**Writing – original draft:** Sohel Ahmed.

**Writing – review & editing:** Sohel Ahmed, Asir John Samuel, Arushi Mishra, Md Saifur Rahman, Md. Ariful Islam, Md. Rashaduzzaman, Shankar Kumar Roy, Rahemun Akter, Mohammad Jahirul Islam.

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
