## [Decision Letter · Decision Letter 0]

11 Jul 2024

PONE-D-24-17302Mobile game addiction and its association with musculoskeletal pain among students: A cross-sectional studyPLOS ONE

Dear Dr. Ahmed,

Thank you for submitting your manuscript to PLOS ONE. After careful consideration, we feel that it has merit but does not fully meet PLOS ONE’s publication criteria as it currently stands. Therefore, we invite you to submit a revised version of the manuscript that addresses the points raised during the review process.

**ACADEMIC EDITOR: Dear author, please justify the queries from the reviewer/s for better understanding on the scientific quality of your research work.** The decision of this manuscript is justified based on PLOS ONE’s publication criteria and not on its novelty or perceived impact.

Please submit your revised manuscript by 10th August 2024 If you will need more time than this to complete your revisions, please reply to this message or contact the journal office at plosone@plos.org. Please include the following items when submitting your revised manuscript:A rebuttal letter that responds to each point raised by the academic editor and reviewer(s). You should upload this letter as a separate file labeled 'Response to Reviewers'.A marked-up copy of your manuscript that highlights changes made to the original version. You should upload this as a separate file labeled 'Revised Manuscript with Track Changes'.An unmarked version of your revised paper without tracked changes. You should upload this as a separate file labeled 'Manuscript'.

We look forward to receiving your revised manuscript.

Kind regards,

Zulkarnain Jaafar

Academic Editor

PLOS ONE

Journal Requirements:

2. We note that your Data Availability Statement is currently as follows: All relevant data are within the manuscript and its Supporting Information file

Reviewers' comments:

Reviewer's Responses to Questions

**Comments to the Author**

1. Is the manuscript technically sound, and do the data support the conclusions?

Reviewer #1: Yes

Reviewer #2: No

2. Has the statistical analysis been performed appropriately and rigorously? 

Reviewer #1: Yes

Reviewer #2: Yes

3. Have the authors made all data underlying the findings in their manuscript fully available?

Reviewer #1: Yes

Reviewer #2: Yes

4. Is the manuscript presented in an intelligible fashion and written in standard English?

Reviewer #1: Yes

Reviewer #2: Yes

5. Review Comments to the Author

Reviewer #1: The article is clearly and objectively written. It presents the main elements suggested by Strobe and provides important information for the field of study.

I suggest removing the lumbar region, as the data (OR 1.78, 95% CI 0.93- 3.40; p = 0.083) show no association.

Reviewer #2: Than you for giving me the chance of reviewing this manuscript. I have 2 questions for you?

1) In this study you searched mobile game addiction but you used a questionnare that assess the internet gaming disorder. there some other questionarres for mobile game addiction or smartphone addiction ( may be this more suitable for your research). why did you sellect this assessment tool?

2) Had the reliablity and validity studies of the questionnaress been done in your language?

6. PLOS authors have the option to publish the peer review history of their article (what does this mean?). If published, this will include your full peer review and any attached files.

Reviewer #1: No

Reviewer #2: No

---

## [Author Response · Author response to Decision Letter 0]

14 Jul 2024

Reply to the reviewer's comments: We express our gratitude to the reviewer for their meticulous and in-depth reading of this manuscript, as well as for their insightful remarks and helpful recommendations, all of which have enabled us to enhance the text's quality.

Reviewer # 1:

Comment No. 1: The article is clearly and objectively written. It presents the main elements suggested by Strobe and provides important information for the field of study.

I suggest removing the lumbar region, as the data (OR 1.78, 95% CI 0.93- 3.40; p = 0.083) show no association.

Reply to comment No 1: We thank the reviewer for the valuable suggestion regarding the removal of lumbar region from the data. However, we intend to inform that, we have included lower back region based on their statistically significant weak association (Chi square = 25.7; p<0.001, Phi-Cramer's v = 0.175; p<0.001) with mobile game addiction from Table 2.

Reviewer # 2: Thank you for giving me the chance of reviewing this manuscript. I have 2 questions for you?

Comment No. 1: In this study you searched mobile game addiction but you used a questionnare that assess the internet gaming disorder. there some other questionarres for mobile game addiction or smartphone addiction (may be this more suitable for your research). why did you select this assessment tool?

Reply to comment No. 1: We thank the thoughtful suggestion. We agree with the reviewer that we have used questionnaire developed and validated for internet gaming disorder. Currently, the majority of mobile phone users (school, college, or university) have smartphones with internet access and spend time playing internet-based mobile games. Hence, it is more appropriate to evaluate the internet gaming addiction as mobile game addiction. For, this reason, we have selected this tool. The above statements are added to the revised manuscript (text) under Materials and Methods, 2.4.2. Game addiction scale, page. No. 7.

Comment No. 2: Had the reliablity and validity studies of the questionnaress been done in your language?

Reply to comment No. 2: Yes, we agree with the reviewer's concerns regarding the use of questionnaire translated and validated in the research participants local language. Unfortunately, due the non-availability of translated version of GAS, we have used the original English language version. As, the student population have sufficient proficiency in English to understand the meaning of each question, we do not find any practical difficulties to use it GAS in the study. We have added the same as one of limitations in the revised manuscript, page no. 15.

---

## [Decision Letter · Decision Letter 1]

30 Jul 2024

Mobile game addiction and its association with musculoskeletal pain among students: A cross-sectional study

PONE-D-24-17302R1

Dear Dr. Ahmed,

We’re pleased to inform you that your manuscript has been judged scientifically suitable for publication and will be formally accepted for publication once it meets all outstanding technical requirements.

Kind regards,

Zulkarnain Jaafar

Academic Editor

PLOS ONE

Additional Editor Comments (optional):

Reviewers' comments:

Reviewer's Responses to Questions

**Comments to the Author**

1. If the authors have adequately addressed your comments raised in a previous round of review and you feel that this manuscript is now acceptable for publication, you may indicate that here to bypass the “Comments to the Author” section, enter your conflict of interest statement in the “Confidential to Editor” section, and submit your "Accept" recommendation.

Reviewer #1: (No Response)

Reviewer #2: All comments have been addressed

2. Is the manuscript technically sound, and do the data support the conclusions?

Reviewer #1: Yes

Reviewer #2: Yes

3. Has the statistical analysis been performed appropriately and rigorously? 

Reviewer #1: Yes

Reviewer #2: Yes

4. Have the authors made all data underlying the findings in their manuscript fully available?

Reviewer #1: Yes

Reviewer #2: Yes

5. Is the manuscript presented in an intelligible fashion and written in standard English?

Reviewer #1: Yes

Reviewer #2: Yes

6. Review Comments to the Author

Reviewer #1: (No Response)

Reviewer #2: Dear authours

Thank you for your convincing explanations and revions.With these changes your manuscript sound more scientific.

Sincerely

7. PLOS authors have the option to publish the peer review history of their article (what does this mean?). If published, this will include your full peer review and any attached files.

Reviewer #1: No

Reviewer #2: No

---

## [Editor Report · Acceptance letter]

14 Aug 2024

PONE-D-24-17302R1 

PLOS ONE

Dear Dr. Ahmed, 

I'm pleased to inform you that your manuscript has been deemed suitable for publication in PLOS ONE. Congratulations! Your manuscript is now being handed over to our production team.

Kind regards, 

on behalf of

Dr. Zulkarnain Jaafar 

Academic Editor

PLOS ONE